# 3D printed graphene-based self-powered strain sensors for smart tires in autonomous vehicles

Deepam Maurya [1,2✉], Seyedmeysam Khaleghian[3], Rammohan Sriramdas[4], Prashant Kumar[1], Ravi Anant Kishore [1,5], Min Gyu Kang[4], Vireshwar Kumar[6,7], Hyun-Cheol Song[8], Seul-Yi Lee[9], Yongke Yan[4], Jung-Min Park[6], Saied Taheri[1,10✉] & Shashank Priya [4✉]

The transition of autonomous vehicles into fleets requires an advanced control system design that relies on continuous feedback from the tires. Smart tires enable continuous monitoring of dynamic parameters by combining strain sensing with traditional tire functions. Here, we provide breakthrough in this direction by demonstrating tire-integrated system that combines direct mask-less 3D printed strain gauges, flexible piezoelectric energy harvester for powering the sensors and secure wireless data transfer electronics, and machine learning for predictive data analysis. Ink of graphene based material was designed to directly print strain sensor for measuring tire-road interactions under varying driving speeds, normal load, and tire pressure. A secure wireless data transfer hardware powered by a piezoelectric patch is implemented to demonstrate self-powered sensing and wireless communication capability. Combined, this study significantly advances the design and fabrication of cost-effective smart tires by demonstrating practical self-powered wireless strain sensing capability.

[1] Department of Mechanical Engineering, Virginia Tech, Blacksburg, VA 24061, USA. [2] Department of Materials Science and Engineering, Virginia Tech, Blacksburg, VA 24061, USA. [3] Department of Engineering Technology, Texas State University, San Marcos, TX 78666, USA. [4] Department of Materials Science and Engineering, Penn State University, University Park, PA 16802, USA. [5] National Renewable Energy Laboratory, 15013 Denver West Pkwy, Golden, CO 80401, USA. [6] Department of Electrical and Computer Engineering, Virginia Tech, Blacksburg, VA 24061, USA. [7] Department of Computer Science and Engineering, Indian Institute of Technology Delhi, New Delhi 110016, India. [8] Center for Electronic Materials, Korea Institute of Science and Technology (KIST), Seoul 02792, Republic of Korea. [9] Institute for Critical Technology and Applied Science (ICTAS), Virginia Tech, Blacksburg, VA 24061, USA. [10] Center for Tire Research (CenTiRe), Virginia Tech, Blacksburg, VA 24061, USA. ✉email: mauryad@vt.edu; staheri@vt.edu; sup103@psu.edu

Strong commercial interest in deployment of autonomous vehicles is driving the development of smart tires that will be required for meeting the safety standards[1,2]. Almost 10 million self-driving cars are expected to be deployed in near future[3], which emphasizes the urgent need for design of precise control and communication subsystem[4]. Smart tires provide the ability to dynamically sense tire-road interaction parameters which are critical towards the design of robust intelligent controls. Ideally, self-powered sensors should be embedded in the tires and measured data should be securely transmitted at high frequencies to enable real-time control. Prior solutions have not been able to meet these requirements, often resulting in cumbersome multistep integration process which adds to the cost and management. Recent research on graphene-based sensors has shown promising results due to its high performance and increased sensitivity[5–11]. Here, we demonstrate that 3D printed sensors coupled with energy harvesting and secure data transmission are cost-effective solutions for smart tires. The cost of a 3D printed sensor in this work was estimated to be roughly 2.7 cents. Graphene-based 3D-printed strain sensors are shown to acquire environmental information and wirelessly transmit the secured data at desired frequencies. The data communication subsystem is shown to be powered by piezoelectric energy harvester that is also embedded within the tire.

Globally, efforts have been made on integrating wireless sensors within tire for measuring dynamic mechanical parameters[12–14]. However, most of these sensors are rigid, require external power and are fabricated using time-consuming multistep processes, which increases the complexity and cost. Comparatively, 3D-printing based manufacturing process not only simplifies processing but also enables direct integration with the tires. Aerosol-based 3D-printing is a versatile process providing ability to fabricate films and patterns of heterogeneous materials[15]. Here, we demonstrate 3D-printed strain sensors for smart tires to measure the tire-road interaction during vehicle movement. The integration of smart tires with autonomous vehicles will enhance their operational safety by providing real-time changes in road friction coefficient. In order to power the sensors, we rely on mechanical energy scavenged from the tire deformation.

Traditionally, the tire pressure monitoring system (TPMS) has been used in-vehicular wireless network, which is employed to warn a driver of any loss in tire pressure. TPMS consists of a pressure sensor, a microcontroller, a radio frequency communication system to wirelessly transmit the data to the vehicle's central processing unit (CPU), and a battery as a power source. The TPMS regularly sends messages containing tire pressure information to CPU. To conserve energy and extend battery life, the rate of measurement and message transmission at the TPMS are traditionally limited to a very low value, e.g., the TPMS transmits only one message per 60 s[16]. Further, due to energy constraints, TPMS does not employ any security mechanism to authenticate the messages sent to the CPU. Hence, the communication between TPMS and CPU can be easily hacked/compromised[17]. The hacked communication means that the CPU may be provided with misleading tire pressure values. This implies a huge risk in the scenario where a driver is alerted with a false reading of low pressure on its dashboard while the car is being driven at a high speed on a highway. In this work, we propose an energy-efficient technique to employ the security mechanisms in a wireless data transfer from the 3D printed strain sensor, which is powered through energy-harvesting system.

## Results
### 3D printing of graphene-based sensor and field testing. The overview of the present work covering printing of self-powered

tire strain sensor with secure data transmission to achieve next generation of autonomous vehicles, is shown in Fig. 1. During the process optimization, first, we printed strain sensors with silver nanoparticle on Kapton film. The process details and related results on these printed sensors are depicted in Supplementary Figs. 1–4. Building upon these initial results, we designed inks for graphene and conducted 3D-printing initial trials, as shown in Supplementary Fig. 5a–c. In order to understand the morphology of the graphene oxide (GO) sheets (used for making 3D-printing ink), bright field transmission electron microscopy (TEM) image analysis was performed (Fig. 2a). Since GO-based ink is stable in water, after printing, GO strain sensor was reduced chemically to reduced graphene oxide (rGO) with enhanced conductivity. After chemical reaction, the color of the printed sensor changed to dark gray from gray Supplementary Fig. 5d, e.

Figure 2b shows the XRD-spectra recorded on 3D printed GO and rGO. The prominent peak at $2\theta \sim 11.11°$ in the GO, corresponds to a significant increase in the interlayer $d$-spacing of 0.7 nm as compared to the graphite ($d$-spacing of 0.335 nm). This implies that the obtained GO was fully exfoliated due to the intercalation of oxygen functional groups. After reduction, weak and broad peak of rGO at $2\theta \sim 25.7$ was observed, which indicates smaller distance between adjunct graphene layers due to restacking of nanostructure. Further to understand structural change in printed sensor, Raman spectra was recorded after chemically reducing 3D-printed GO sensors, as shown in Fig. 2c. The Raman spectra confirmed structural changes in reduced graphene with respect to the GO. The D band can be attributed to in-plane $A_{1g}$ (LA) zone-edge mode. The D band, located near $1367 \text{ cm}^{-1}$ originates from a defect induced breathing mode of $sp^2$ rings. The G band at $1598 \text{ cm}^{-1}$ can be attributed to the first order scattering of the $E_{2g}$ phonon of $sp^2$ hybridized C atoms. The D band intensity is related to the size of the in-plane $sp^2$ domains[18]. The ratio of the relative intensity of D and G peaks i.e. ($I_D/I_G$) can be used to measure the degree of disorder and is

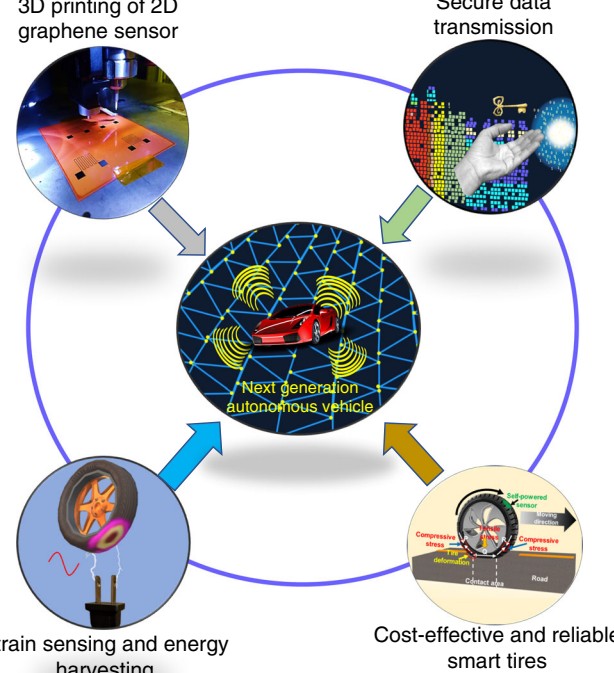

**Fig. 1 Overview of the present work.** Smart tires with innovative 3D printed graphene-based strain sensors, secure data transfer, and strain energy harvesting for next generation of autonomous vehicles.

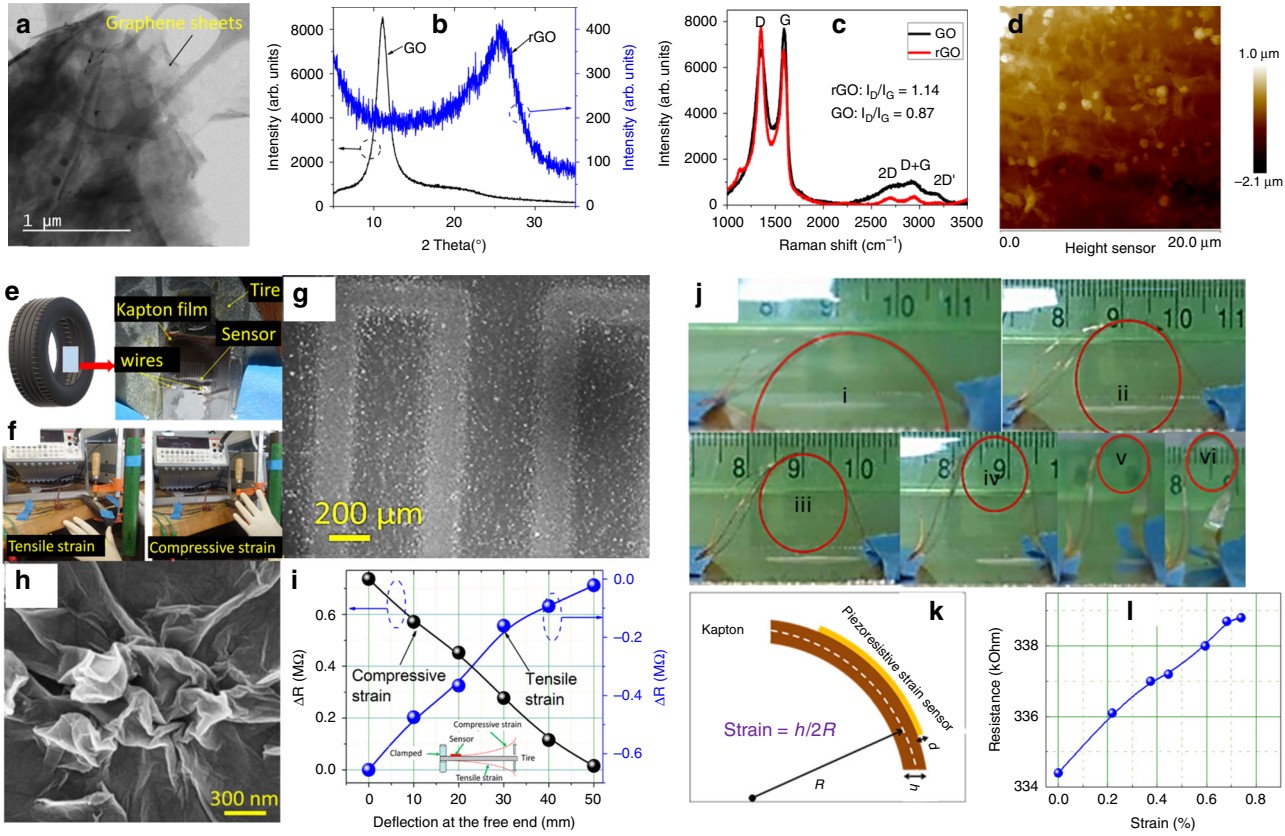

**Fig. 2 3D printed sensor characterization. a** Bright field TEM images of GO sheets used for 3D printing. To increase conductivity for sensor sensitivity, these GO sheets were reduced chemically after printing. **b**, **c** XRD and Raman spectra of GO and rGO delineating structural differences between GO and rGO. **d** AFM images (20 µm × 20 µm) of the 3D printed rGO surface without any major roughness issue. The height scale is given adjacent to **d**. **e** 3D printed graphene-based piezoresistive sensor on a piece of tire. The tire image courtesy of Free3D. **f** Lab measurement setup for measuring change is resistance under tensile and compressive strain. **g**, **h** The microstructure of the 3D printed graphene sensor. Please note wrinkled microstructure of graphene sheets, which allows high flexibility. **i** Change in resistance under compressive and tensile strain. **j** Bending of the strain sensor printed on Kapton with different radius of curvatures. The red circles are drawn to overlap the curvature of Kapton due to bending in **j**. These circles also act as a guide to eyes. **k** Schematic showing strain measurement from bending. **l** Resistance versus strain plot from the bending of the sensor.

inversely proportional to the average size of the $sp^2$ clusters. In our case, the intensity ratio ($I_d/I_G$) of the reduced graphene was found to be higher than that of the GO. This indicates generation of new $sp^2$ domains and increased disorder[19]. The 2D band located at higher frequency originates from the double-resonance process and is highly sensitive to the number of graphene layers (usually only less than four layers). For a single layer graphene, this mode is generally sharp and splits into several layers with the addition of more layers. Another mode (D + G) is a combination mode originating from the disorder in the system. rGO has been found to exhibit long $sp^2$–$sp^2$ conjugated carbon chain, which facilitates electron transport increasing electrical conduction[20]. These results indicate structural differences between GO and rGO, and confirm that rGO based sensor will have favorable response to strain through corresponding changes in conductivity/resistance. After controlling the process variable such as gas flow, speed, and viscosity, we were able to achieve GO sensors with homogenous microstructure (Fig. 2d–g). The final graphene-based printed sensor was obtained using five passes with total thickness of ~10 µm.

Next, we integrated these graphene-based piezoresistive sensors on a piece of tire and performed measurements using lab setup as shown in Fig. 2e, f. Please note the change in resistance under compressive and tensile strain (Fig. 2i). The wrinkled microstructure (Fig. 2h) of graphene sheets allows withstanding of large deformations without damaging the sensors. We measured

change in resistance of these rGO sensors on flexible Kapton films under bending with different radius of curvature, as shown in Fig. 2j. The thickness of the substrate and the radius of curvature during the bending were used to calculate the strain (Fig. 2k). Figure 2e shows resistance versus strain plot for the bending experiment. Please note an almost linear change in the resistance up to ~0.7% strain. We also studied time dependent change in the resistance of the 3D printed rGO based sensor under tensile strain (Supplementary Fig. 6).

To demonstrate the feasibility of the printed sensor, the 3D printed graphene-based sensor was integrated onto a tire (Goodyear, model # P245/70R17) of a mobile test rig (Fig. 3a) and performance was measured in real environment driving conditions (Fig. 3b). A video of the experimental setup is shown in the Supplementary Movie 1. For collecting data from the tire sensor, the test rig was operated between parking A and parking B (Fig. 3b) which are ~300 m apart. In one of the scenario's, the vehicle was operated from A–B–A and thereby completing 600 m under one condition. The total distance covered during this data collection was more than 10 miles, which clearly indicates robustness and reliability of the 3D printed sensor.

There are three zones of strain in a tire during driving conditions; two compressive strain zones (zone P and Q), and one tensile strain zone (zone R), as shown in Fig. 3c. Figure 3d shows the waveform due to tire movement on the ground. The change in the voltage was measured due to the varying resistance of the

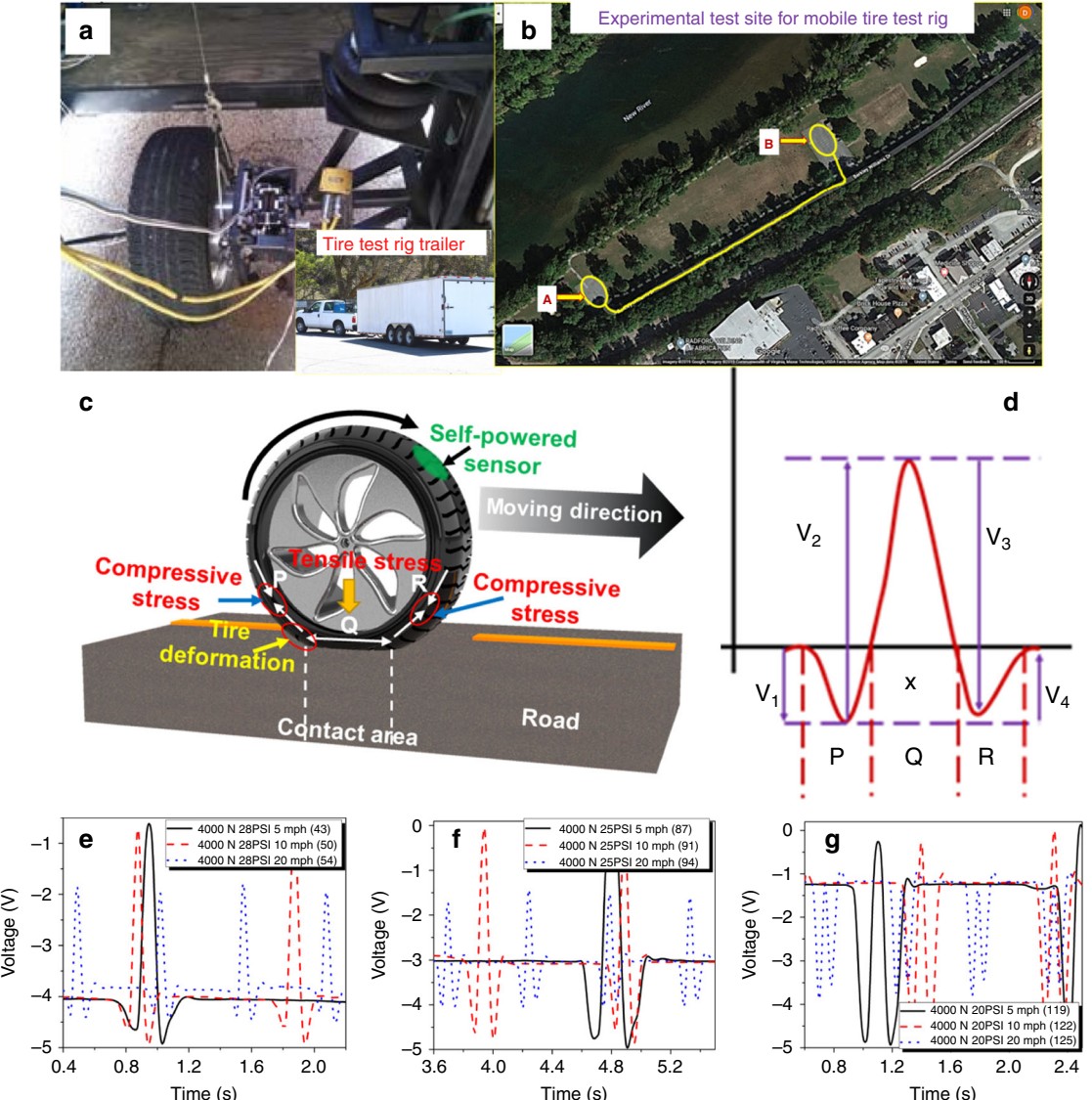

**Fig. 3 Field test setup and results of the 3D printed tire sensor. a** Mobile tire test rig for testing of the sensors in a real environment. The inset of the **a** shows that the mobile tire test rig was attached to a towing vehicle. **b** Satellite map (Courtesy google map) of the experimental test site (Bisset Park, Radford, VA) of the smart tire in a test rig. The measurements were performed on smart tire, while operating test rig between parking A and parking B. **c**, **d** Schematics of the tire deformation with different types of strain and corresponding waveform. The varying voltage output across the 3D printed piezoresistive sensor at 4 kN normal load and three different speeds (5, 10, and 20 mph) for **e** 28 psi, **f** 25 psi, and **g** 20 psi.

printed sensor due to tire deformation. To understand the effect of various parameters, the measurements were performed at different speeds, normal loads, and tire pressures, as shown in Supplementary Table 1. During real environment field test on the 3D printed sensor, the sensor was subjected to a large number of cycles ~8031 (Supplementary Table 2). Figure 3e–g show voltage waveforms due to tire movement under a normal load of 4000 N at three different speeds (5, 10, and 20 mph) and three different tire pressures (20, 25, and 28 psi). Please note the change in voltage waveform due to change in various factors. The bias in the voltage output at each tire pressure is attributed to the shift in the mean resistance of the sensor with the ambient pressure. For a given normal load and tire pressure of 28 psi, the peak in the voltage output was found to be lower for higher speed of 20 mph, which could be attributed to the decreased tensile strain zone. The variations in output voltage with speed were almost similar for 25 psi tire pressure. Interestingly, for 20 psi tire pressure and 4000 N normal load, the highest peak in the voltage output corresponded

to the compressive strain zones. However, the peak height due to the compressive strain was found to be lower for higher speeds. These results indicate that the peak height due to both tensile and compressive strains decreases with increasing speed under given conditions. The measurement results at two different normal loads of 2000 and 3000 N are depicted in Supplementary Fig. 7. Next, theoretical calculations were performed in order to understand the development of strain in a moving tire and its interaction with road.

**Tire modeling**. The strain in the tire can be determined by assuming the tire as a ring suspended by radial and tangential springs[21]. The radial and tangential stiffness of the tire were determined by matching the measured tire frequencies. Here, the radial and tangential displacements in the tire are denoted by $w(\theta, t)$ and $v(\theta, t)$, respectively. The tangential stiffness, radial stiffness, equivalent flexural rigidity, and density of the tire are denoted by $k_v$, $k_w$, $EI$, and $\rho$, respectively. The governing equation of motion

for a rotating inextensible tire with radius $r$, width $b$, thickness $t$, and internal pressure $p$ is derived using the Hamilton's principle[22]. The equation of motion for a nonrotating tire to determine the natural frequencies is obtained by ignoring the tire rotation and tire loading as:

$$\frac{EI}{r^4}\left(v^6 + 2v^4 + v''\right) - \frac{pb}{r}\left(v^4 + v''\right) + k_w v'' - k_v v - \rho bt(\ddot{v} - \ddot{v}'') = 0. \tag{1}$$

The frequencies were estimated by assuming a series solution for the in-plane deflection, $v$. We compared the first three experimental frequencies to match with the predicted frequencies by estimating $EI$, $k_w$, and $k_v$. The series solution in terms of the generalized coordinates $a(t)$ and $b(t)$ for $v$ is assumed to be of the form[23]:

$$v = \sum_{n=1}^{\infty} a_n(t)\cos(n\theta) + b_n(t)\sin(n\theta). \tag{2}$$

The above equation is substituted into Eq. (1) for a rotating tire subjected to an applied radial force $q_w$ and rotation rate $\Omega$. The partial differential equation is transformed into the following two ordinary differential equations:

$$\begin{aligned}
m_n\ddot{a}_n + c_n\dot{a}_n - g_n\dot{b}_n + k_n a_n = \xi_n, \\
m_n\ddot{b}_n + c_n\dot{b}_n + g_n\dot{a}_n + k_n b_n = \eta_n,
\end{aligned} \tag{3}$$

where,

$$m_n = \rho bt(1 + n^2), \quad c_n = 2\zeta\sqrt{m_n k_n}, \quad g_n = 4\rho btn\Omega,$$
$$\xi_n = \frac{1}{\pi}\int_0^{2\pi} q'_w \cos n\theta d\theta, \quad \eta_n = \frac{1}{\pi}\int_0^{2\pi} q'_w \sin n\theta d\theta,$$
$$k_n = \frac{EI}{r^4}n^2(1-n^2)^2 - \frac{pb}{r}n^2(1-n^2) + \rho bt\Omega^2(n^2-3)n^2 + k_w n^2 + k_v. \tag{4}$$

After determining the in-plane deflection, the strain at a height of $z_0$ from the neutral plane of the tire ring was determined using the following equation:[24]

$$S_{11} = \frac{1}{2r^2}(v + v'')^2 + \frac{z_0}{r^2}(v' + v'''). \tag{5}$$

The in-plane strain given by Eq. (5) was derived by assuming the inextensible condition. Furthermore, it is assumed that the graphene sensor weight and stiffness have negligible effect on the tire dynamics. A complete transfer of strain from tire to the graphene sensor is assumed due to the negligible damping occurring at the interface between tire and sensor. Thus, the strain generated due to tire interaction with road is independent of the material properties of the graphene sensor. The strain estimated by Eq. (5) is taken to be the strain experienced by the graphene sensor. The height $z_0$ from the neutral layer towards the inner surface of the tire ring corresponded to the sensor mounting location. The strain at the sensor location for different speeds, tire pressure and load was estimated and compared with the data recorded experimentally from the graphene sensor.

**Simulation**. The model number of the tire selected for the study is P245/65R17 105S. The geometry of the tire is obtained from the tire specifications. Based on the experimentally observed frequencies, the flexural rigidity, radial and tangential stiffness were estimated to be $63.6\,\mathrm{Nm^2}$, $1562\,\mathrm{kNm^{-2}}$, and $2212\,\mathrm{kNm^{-2}}$. A damping ratio of 0.1 was assumed for all the modes of vibration[22,25]. The strain in the tire was estimated at different speeds, loads and tire pressures using Eq. (5). The selected speeds were 5, 10, and 20 mph, and the tire pressures were 20, 25, and 28 psi. The dependence of strains at 2000, 3000, and 4000 N was also estimated. The strain as a function of time for three speeds at 3000 N load and 28 psi pressure is shown in Fig. 4a. It was

observed that the amplitude of the strain marginally changes with speed. However, at a given speed and pressure, the peak value of strain was found to increase with load, as shown in Fig. 4b. The strain as a function of time in the case of varying pressures is shown in Fig. 4c. These results indicated higher strain under lower tire pressure, as observed experimentally. A similar observation could be noted in the radial displacement of the tire. In experiments, the voltage is measured across an equivalent resistance connected in series with the graphene sensor and a DC power supply. The comparative analysis of modeling and experimental results (Supplementary Fig. 8) clearly indicate that the simulated strain response follows the same trend as the experimental results shown in Fig. 3e. As shown in Fig. 4d, the peak displacement increases with decreasing tire pressure. For three pressure values, the radial displacement as a function of angular position is shown in Fig. 4d. A representative plot of the deformation of tire under a given load of 3000 N and a tire pressure of 28 psi is shown in Fig. 4e. The model can be further improved to predict the variation in strain with different speeds by considering the transient behavior of the deformation. The steady state response of the deformation clearly predicts the dependence of strains on the tire pressure and load, corroborating the experimental observations. Moreover, the amplitude of the sensor output is a direct representative of the strain in the selected pressure ranges. It is observed that the developed graphene sensor is capable of measuring strains on the order of 3500–6000 μ. Hence, our 3D printed strain sensor is a promising solution for sensing tire strain and monitoring the tire health. Next, to demonstrate the application of these 3D printed sensors in autonomous vehicles, we developed a machine learning algorithm to estimate the tire pressure condition.

**Machine learning**. As shown in Fig. 3c, four parameters can be defined on the output voltage waveform in one tire revolution. The time derivative of the signal can be used to estimate the contact patch length and since the contact patch length can be correlated to tire normal load, tire normal load at a specific tire pressure can be calculated[26]. Figure 5a, b show the experimental output signals of the printed piezoresistive sensor and its time derivative at 28 psi and 5 mph under normal load of 2000 N. Figure 5c–f depict change in the magnitude of parameters V1 and V4 for different tire normal loads and inflation pressures, respectively, for a selected batch of data. Both of the parameters V1 and V4 were found to increase significantly with the reduction in tire pressure or with the increase in the tire normal load. However, these parameters remained nearly unchanged with tire velocity. Therefore, these parameters can be used to estimate tire pressure.

Figure 6a shows a schematic of the machine learning algorithm. The tire pressure was estimated using the tire normal load, longitudinal velocity and the parameter V1. The experimental data of 1023 tire revolutions collected using the portable tire-testing trailer was used to develop the neural network (NN) based pressure monitoring algorithm. Roughly 70% of the data was randomly selected and used for training, 15% for validation, and 15% for testing. A two-layer feed-forward NN was used with sigmoid hidden neurons that uses ten neurons in its hidden layer. The fitting performance for the training, validation and testing is shown in Fig. 6b–e. The correlation coefficient was found to be higher than 0.96 for all the cases. The error histogram for the tire pressure monitoring algorithm is shown in Fig. 7. The error was defined as a difference between the estimated tire pressure and the reference one. The majority of the data points were found to be within a close range of zero error line, which exhibited the viability of the printed sensor in monitoring the tire pressure successfully.

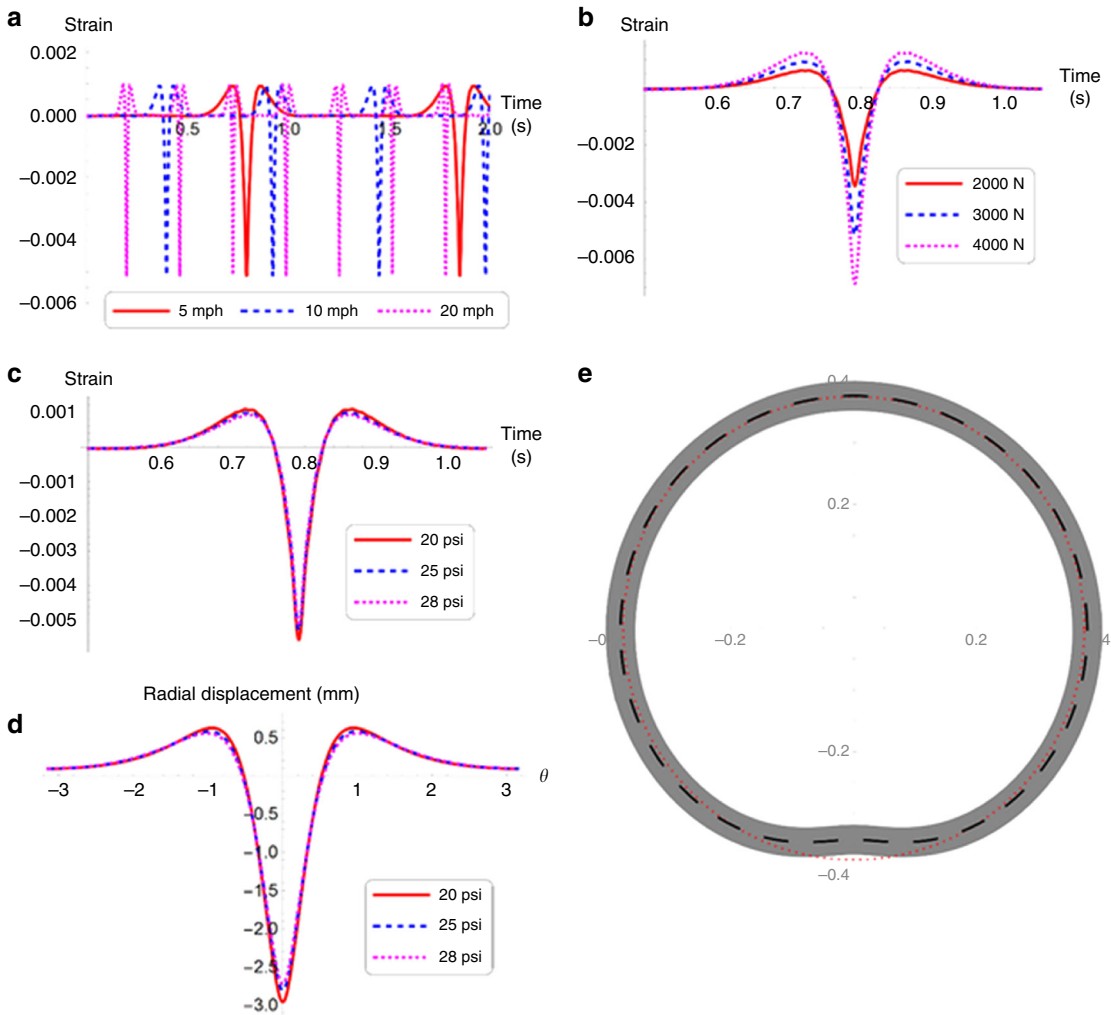

**Fig. 4 Tire modeling results. a** The strain as a function of time for three speeds is estimated using Eq. (5) for a load of 3000 N and pressure of 28 psi. It can be observed that the amplitude of the strain remains the same with speed. **b** The simulated strain in the tire as a function of time for three loads is shown for a speed of 5 mph, and pressure of 28 psi. **c** The simulation results for strain as a function of time for three pressures at a speed of 5 mph and a load of 3000 N. **d** The simulated radial displacement as a function of angular position for three tire pressures for a load of 3000 N and speed of 5 mph. It can be observed that the displacement and hence the strain increases with decrease in the tire pressure for a given load. **e** A representative plot of the simulation results for tire deformation (m) at any given speed. The displacement shown corresponds to that at 5 mph, 3000 N force and 28 psi pressure with a five times amplification.

**Power management and secure data transfer**. We demonstrated the wireless data transfer between a commercial wireless sensor (which was powered by a piezoelectric patch mounted on part of a tire) and a corresponding mobile app. Figure 8a shows the setup for energy generation using a Polyvinylidene fluoride (PVDF) piezoelectric patch (3 cm × 7 cm) mounted on a piece of tire, which was excited with a shaker at a selected frequency and amplitude. The measurement frequencies were chosen based on the certain angular velocities of a tire with a given diameter. The energy generated was stored in a capacitor (5 μF) for powering the wireless sensor system (MIDASCON), which was mounted on a tire of a cart rotating on a walk-mill, as shown in Fig. 8b. The power versus load resistance plot at 13 Hz is depicted in the inset of Fig. 8b. The tire was heated with a heat gun resulting in temperature rise of the tire and reduction in the humidity (Supplementary Movie 2). This information was wirelessly transferred to a cell phone as can be seen in the zoomed screen of the cell phone app (Fig. 8b). We successfully demonstrated powering of the data transfer using energy generated through a piezoelectric polymer mounted onto a tire.

Traditionally, to secure each transmitted message containing sensor measurements, the sensor in a smart tire needs to run a security algorithm twice: firstly, to generate the authentication information, and then secondly to encrypt the message[27]. We note that in a normal-event scenario, the sensor measurements (e.g., tire pressure) can remain the same for hours, but in a rare-event scenario (e.g., puncture), the measurements can change rapidly (e.g., every second). Since the message transmission mechanism from the sensor to the vehicle's CPU is designed to support the rare-event scenario, the sensor measurements in a normal-event scenario may remain the same for a large number of consecutively transmitted messages. Exploiting this fact, we propose the following novel mechanism for securing the sensor's messages. When the sensor computes the authentication information for a message by running the security algorithm, it stores the message with the sensor's measurements and the authentication information in a database. Thereafter, whenever the sensor detects that the measurements to be transmitted in the current message has already been transmitted in one of the previous messages, it simply utilizes the corresponding

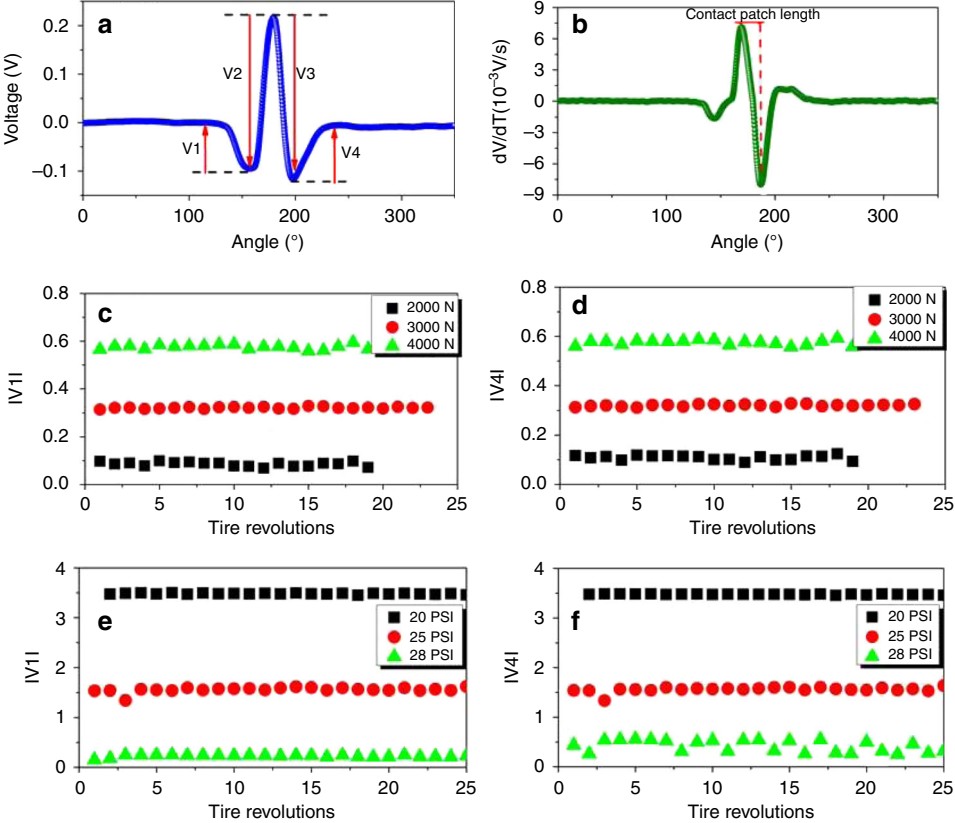

**Fig. 5 Experimental results of the printed tire sensor. a** The voltage output signal in a full tire revolution for the tire with pressure of 28 psi, velocity of 5 mph, and the normal load of 2000 N. **b** The time derivative of voltage signal. The effects of tire normal load on (tire pressure = 28 psi, velocity = 5 mph). **c** The magnitude of parameter V1. **d** The magnitude of parameter V4. The effects of tire inflation pressure on (tire load = 2000 N, velocity = 20 mph). **e** The magnitude of parameter V1. **f** The magnitude of parameter V4.

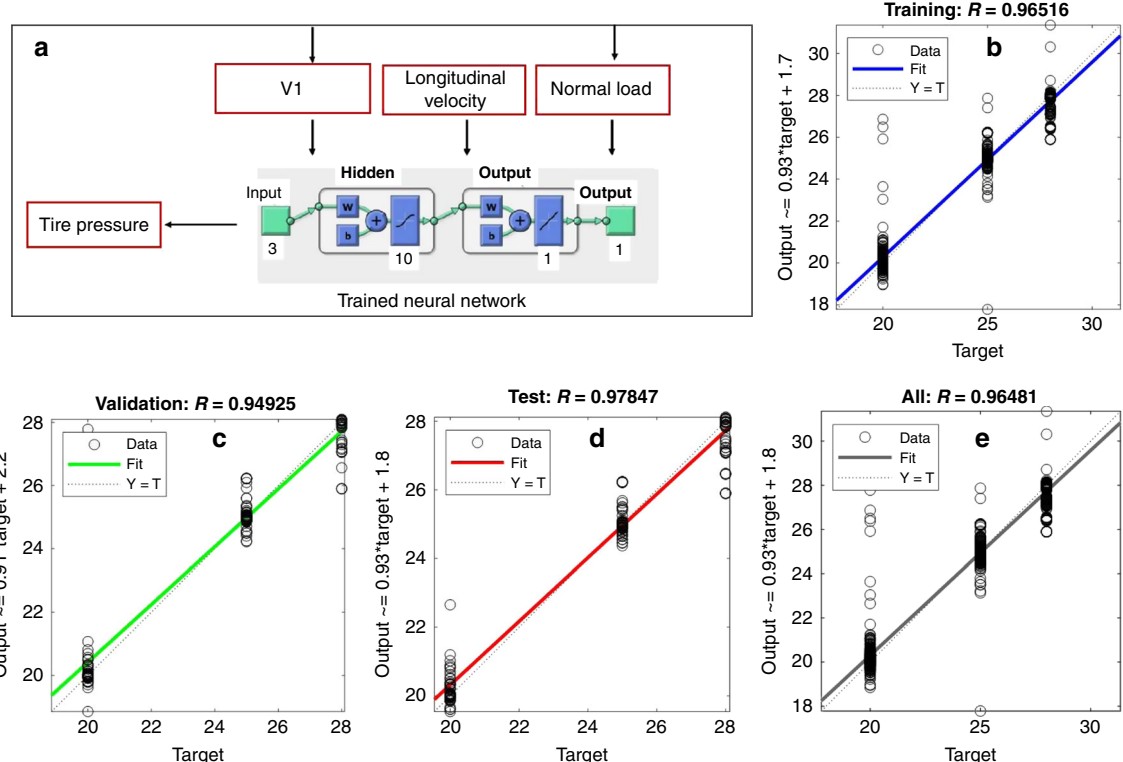

**Fig. 6 Machine learning algorithm and fitting performance. a** The schematic of tire pressure monitoring algorithm. **b–e** The fitting performance for training, testing, and validation.

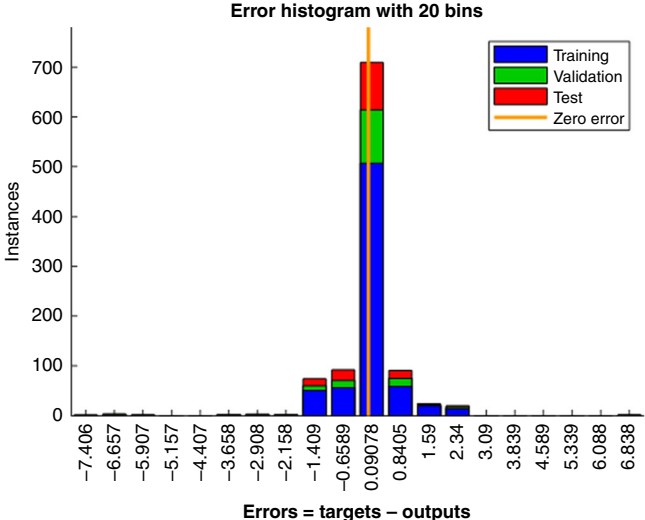

**Fig. 7 Machine learning performance in tire pressure monitoring.** The error histogram for the tire pressure monitoring algorithm.

authentication information from the database. Therefore, for each repeated message, the sensor needs to run only one instance of the security algorithm to encrypt the message.

We evaluated the performance of our proposed mechanism using the energy measurements for a low-power automotive grade wireless sensor utilized for energy-harvesting applications[15]. In Fig. 8c, we present our results corresponding to three scenarios: the cost of measurement and transmission of message without security, that with the conventional security mechanism, and that with the proposed security mechanism. We observe that the proposed idea of reusing the previously computed authentication information significantly reduces the computation overhead of security. Our results also demonstrate that 3D printed sensors in combination with piezoelectric energy harvesters can meet the requirements for secure wireless sensor network on a tire platform.

## Discussion

In summary, we developed a printing process for piezoresistive sensors on varying substrates using aerosol deposition (AD) method. To achieve high flexibility, we fabricated graphene ink

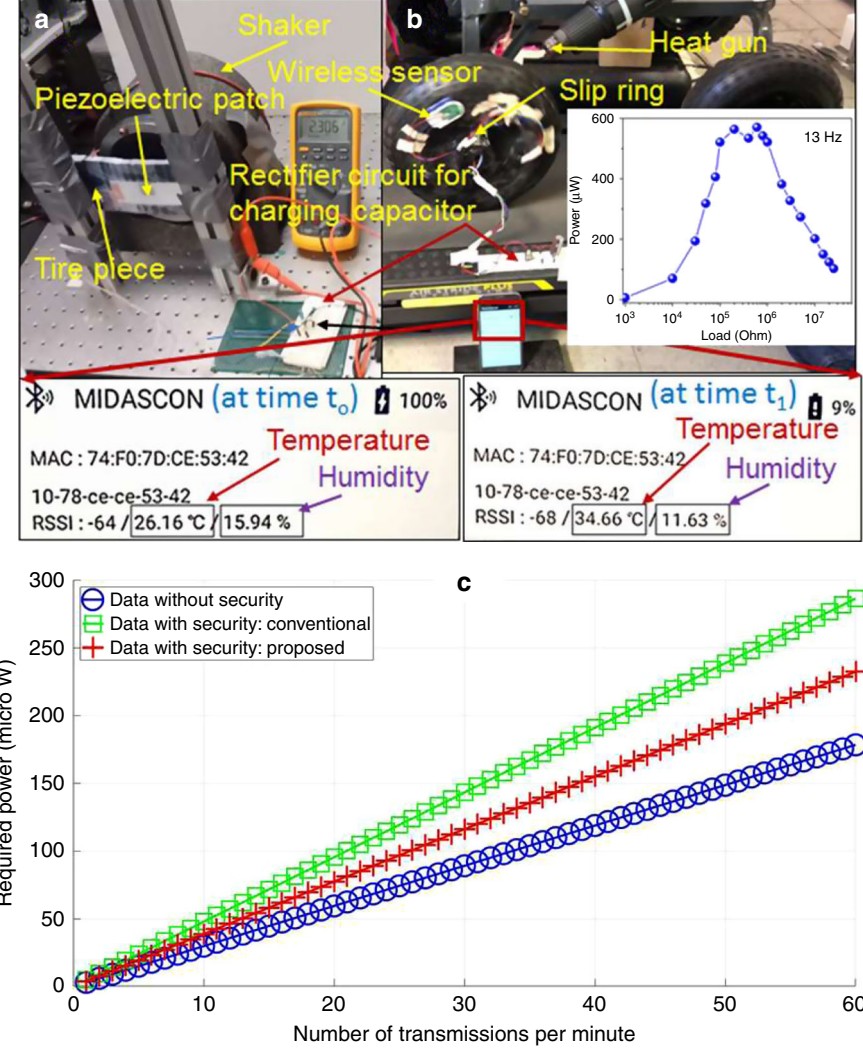

**Fig. 8 Energy harvesting to power secure data transmission. a** Photograph of the piezoelectric patch mounted on a piece of tire for energy generation. **b** Demonstration of the tire temperature and humidity sensing using a wireless sensor powered by the energy stored in a capacitor. This wireless sensor transferred information wirelessly to a mobile phone screen through a commercial app. The magnified view of the mobile phone screen, at two different conditions, is shown in the bottom of **b**. The inset of **b** shows peak power versus load resistance plot obtained from the system shown in **a**. **c** Required power versus number of transmissions per minute with security, without security (conventional), and our proposed method.

for fabricating the sensors. The wrinkled microstructure of graphene allowed withstanding large deformations without failure. These 3D-printed graphene sensors were integrated into an actual tire of a mobile test rig to measure voltage waveforms due to strain generated in a tire during motion. We observed a substantial change in the output waveform due to changes in various parameters like normal load, speed, and tire pressure. Theoretical calculations were performed to successfully model and simulate the experimental results on 3D printed sensors mounted inside a tire. To demonstrate practical feasibility of the 3D printed tire sensor, we developed a machine learning algorithm for estimating tire pressure. Most of the data points were found to be within a close range of zero error line, which exhibited the viability of the printed sensor in monitoring the tire pressure successfully. Further, we demonstrated wireless data transfer by harvesting tire strain energy and developed an energy-efficient technique to employ the secure wireless data transfer. We believe our results will pave the path for next generation of smart tires for autonomous vehicles.

## Methods

**Ink preparation and 3D printing**. AD[28] was used to print piezoresistive sensors. Using this process, one can easily print various architectures with line widths in the range of 10–100 μm. For printing sensors, graphene, and silver nanoparticle-based ink was used. For printing silver nanoparticles, commercial ink was used. For printing graphene-based strain sensors, we developed our own chemistry. Graphite powder (particle size <20 μm, Sigma-Aldrich Co.) was used as a starting material and Hummers method was used to synthesize GO[29]. Details are as follows; 1 g of graphite was immersed in a mixture of sulfuric acid (98%, 120 mL) and phosphoric acid (85%, 15 mL) in an ice-water bath to keep the temperature of the reactor cool around 5 °C. Potassium permanganate (6 g) as an oxidizing agent was slowly added to the mixture and the reaction was maintained at 50 °C for 24 h. After cooling to room temperature, deionized water (200 mL) and hydrogen peroxide (30%, 3 mL) was added to the mixture until it gets a brilliant yellow suspension. The yellowish GO suspension was washed with hydrochloric acid (5%) to remove unreacted metal residues and neutralized with deionized water several times. The obtained GO solution was further exfoliated using a probe-type ultra-sonication and separated by centrifugation to remove unreacted graphite. The GO thus obtained was diluted with water for the aerosol-based 3D printing (Aerosol Jet® printer). For printing process pneumatic atomizer was used. The printed GO film was converted to rGO using hydriodic acid (57%, Sigma-Aldrich). Using present 3D-printing method, sensors can be directly printed on various kinds of substrates (Supplementary Fig. 9).

**Materials characterization**. The XRD-spectra for GO and rGO were collected using X-ray diffractometer (D8 Advance, Bruker). Raman spectra were collected using a Jobin-Yvon LabRam HR 800 high-resolution Raman spectrometer with a laser radiation of 514.5 nm from a Coherent Innova 99 argon source. The laser beam was focused to an area of ~2-μm diameter using a Raman microprobe with a ×50 objective. The morphology of the printed sensor was analyzed using a field emission scanning electron microscope (LEO (Zeiss) 1550). An atomic force microscope (Bruker Dimension Icon, Billerica, MA) was used to record surface morphology. The TEM images were recorded using a FEI Titan 300 microscope.

**Piezoresistive sensor**. The piezoresistive sensors are based on the change in the resistance, which can be given by the Taylor series expansion[30]:

$$\Delta R = \frac{\partial R}{\partial \rho}\Delta\rho + \frac{\partial R}{\partial L}\Delta L + \frac{\partial R}{\partial A}\Delta A + \text{higher order trems.} \quad (6)$$

Neglecting higher order terms and dividing by $R$,

$$\frac{\Delta R}{R} = \frac{\Delta R}{\rho} + \frac{\Delta L}{L} - \frac{\Delta A}{A}, \quad (7)$$

where, $R$ is resistance, $L$ is length, $\rho$ is resistivity, and $A$ is cross-sectional area of the sensor. First-term in the above equation represents the change in the resistance due to changes in resistivity. The 2nd term represents the change in resistance due to change in dimension.

**Electrical characterization and field test of the 3D printed sensor**. In a lab environment, the resistance change of the printed sensor was measured using a multimeter. For the field test, the 3D printed strain sensor was integrated with a tire of a mobile quarter car test rig installed in a trailer, which is towed by a truck. This test rig is equipped with the setup for changing normal load and tire pressure. The normal load on the tire was varied using a pneumatic pressure transducer. In present work, a constant voltage 9V was applied across 3D printed graphene sensor and the

variation in voltage due to change in its resistance, under external stimuli, was measured. LABVIEW was used for data acquisition during the field test. The measurements were performed at different normal loads, tire pressure, and speeds. For powering wireless sensor, we used commercial PVDF-based piezoelectric materials (Kureha Corporation) integrated on a piece of a tire. This system was excited with a shaker and the resulting energy was stored in a capacitor to power the wireless sensor for transferring the information related to temperature and humidity.

## Data availability
The data related to the findings of this paper can be requested from the corresponding authors upon a reasonable request.

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

## Acknowledgements

D.M. and S.P. gratefully acknowledge financial support from National Science Foundation through I/UCRC: Center for Energy-Harvesting Materials and Systems (CEHMS). P.K. acknowledges the financial support through Office of Naval Research through grant number N000141712520. Y.Y. acknowledges the financial support through the National Science Foundation through grant number ECCS-1832865. M.G.K. acknowledges the support through the Air Force Office of Scientific Research under award number FA9550-17-1-0341. R.S. acknowledges the support through Office of Naval Research through grant number N000141613043. H.C.S. acknowledges support through the NSF-CREST grant number HRD-1547771, the Energy Technology Development Project (KETEP) grant funded by the Ministry of Trade, Industry and Energy, Republic of Korea (2018201010636A), and the National Research Council of Science & Technology (NST) grant by the Korea government (MSIP) (No. CAP-17-04-KRISS). R.A.K. acknowledges the financial support through DARPA MATRIX program.

## Author contributions

D.M., S.P., and S.T. conceived the idea. D.M. designed experiments, developed printing process, and led manuscript writing. S.K. helped with the field test experiment design and machine learning. R.S. helped with the theoretical modeling. P.K. helped with the tire sensor data collection, energy-harvesting experiments, and data analysis. R.A.K. helped with the field test data acquisition and analysis. M.G.K. helped with graphene sensor integration, ink development, and tire energy-harvesting experiments. V.K. and J.M.P. helped with the power management and secure data transmission. H.C.S. helped in designing sensor architecture. S.Y.L. helped with the synthesis of the GO paste for ink fabrication. Y.Y. helped with the sensor design. S.P. and S.T. supervised the overall research work. All the authors discussed the results and contributed to the manuscript.

## Competing interests

The authors declare no competing interests.
