## [Peer Review File · Nature Communications]

Reviewers' comments:

Reviewer #1 (Remarks to the Author):

Overall, the experiments are well-performed and except for some comments that are discussed below, the data and conclusions are reasonable. I would recommend the manuscript for publication in Nature Communications, provided the authors reasonably address the following point.

The main concerns about this manuscript is that it is still unclear the proposed strain sensor, energy harvesting, and wireless transmission can withstand the harsh tire operating conditions, namely abrupt very large strain, high cyclic loading, high strain rate (corresponding to the driving speed of 120 km/h), and high temperature during driving.

The authors should add the some explanation about the limitation of the proposed system so that the reader can understand and utilized the knowledge obtained in this study more effectively.

Reviewer #2 (Remarks to the Author):

Overall comments

This paper has several interesting components on material, manufacturing, modeling, simulation, and machine learning for the realization of self-powered sensors for tires in order to contribute to the driverless car effort. It is timely and the paper posed important problem. The main point emphasized here was the ability to 3D print graphene as a strain sensor and self-powering (combined effort of the two aspects) for the tire and evaluation of the system in a field test. Authors have used aerosol printing technique and they have prepared their own ink in order to print the graphene, with the detail chemistry provided in the manuscript. It was also stated as cost-effective solution.

There are some areas authors might consider to improve the technical detail as well as the literature, to make the paper compelling as well as address the knowledge gap in the literature in convincing manner. For example, some closely related papers on graphene sensing and fabrication are listed below.

[1] Cheng, Yin, Ranran Wang, Jing Sun, and Lian Gao. "A stretchable and highly sensitive graphene-based fiber for sensing tensile strain, bending, and torsion." *Advanced materials* 27, no. 45 (2015): 7365-7371.

[2] Yan, Chaoyi, Jiangxin Wang, Wenbin Kang, Mengqi Cui, Xu Wang, Ce Yao Foo, Kenji Jianzhi Chee, and Pooi See Lee. "Highly stretchable piezoresistive graphene-nanocellulose nanopaper for strain sensors." *Advanced materials* 26, no. 13 (2014): 2022-2027.

Related works in 3D printing of graphene.

[3] Burela, Ramesh Gupta, Jagath Narayana Kamineni, and Dineshkumar Harursampath. "Multifunctional polymer composites for 3D and 4D printing." In *3D and 4D Printing of Polymer Nanocomposite Materials*, pp. 231-257. Elsevier, 2020.

[4] Kamran, Urooj, Young-Jung Heo, Ji Won Lee, and Soo-Jin Park. "Functionalized carbon materials for electronic devices: a review." *Micromachines* 10, no. 4 (2019): 234.

[5] Toyserkani, Ehsan, and Elahe Jabari. "Method and apparatus for aerosol-based three-dimensional (3D) printing of flexible graphene electronic devices." U.S. Patent 10,400,119, issued September 3, 2019.

[6] Kalsoom, Umme, Pavel N. Nesterenko, and Brett Paull. "Recent developments in 3D printable

composite materials." RSC advances 6, no. 65 (2016): 60355-60371.

[7] Wei, Xiaojun, Dong Li, Wei Jiang, Zheming Gu, Xiaojuan Wang, Zengxing Zhang, and Zhengzong Sun. "3D printable graphene composite." Scientific reports 5 (2015): 11181.

The paper is great but there are some issues that are not clear to this reviewer. The self-powered aspect is very important and authors showed using a piezoelectric system. However, how is the graphene base sensor (piezoresistive material) connected with the flexible piezoelectric energy harvester (PVDF)? Some information such as size of the harvester is also missing.

Some other comments on the theory.

It looks like Fig.4 is simulation results. If so, please put simulation results in the caption of Fig.4. How are the material properties of the 3D printed graphene included in the simulation? Not clear. It looks like Eq.5 doesn't have the material property of graphene. Please comment on the manuscript.

Some areas need clarification or discussion. For example, In Fig. 3(e)-(g), why the outputs have different magnitude of bias? This is not discussed.

On the modeling, I think the purpose of the modeling is to compare with the experiment. The best way to show is the error between the simulation model and the experimental results in a plot (if possible). If not, in a text, explaining error magnitudes. Another question: From where Eq.1 came from? Where is Eq.5 used? These issues need clarification.

It was shown that the 3D printed graphene can measure strain 3500 to 6000 microstrain, and it has four layers. What was the thickness of each layer and how about the total 3D printed part thickness?

Machine learning: The results in Fig.5 looks great. It can be easily seen the effects of load and pressure. It was stated that "experimental data of 1023 tire revolutions collected using the portable tire-testing trailer" What was the data size, training time, conditions, and hyper parameters?

Lastly, the cost-effective solution needs some substantial information, quantifying will make it stronger.

Some of the figures are low resolution, less clear and need replacement.

Overall, this paper presented a 3D printed graphene sensor for tires of autonomous vehicles, which has been validated in a field test showing functionality over a range of parameters such as pressure and normal load. The entire work is a very good example of cyber physical system, as it has data security and communication, sensing, and material innovation, monitoring, and machine learning. The issue of realizing usable sensors like graphene that is reliable and feasible solution for autonomous vehicles is an important problem.

Reviewer #3 (Remarks to the Author):

This paper presents an experimental study that investigated separately the different components that would comprise of a smart tire. A main focus of this study was to present a 3D-printed graphene strain gage. The reviewer found several issues with this manuscript, and the comments are as follows:

- The abstract states that dynamic parameters such as road condition, friction, slip, pressure, and strain distributions are required. This study does not demonstrate that any of these parameters could be directly measured.
- Complete characterization of the graphene-based sensor is basically nonexistent in this paper.

The only result presented was the sensor mounted on a piece of tire and loaded. Sensor properties, as well as their cyclic response and performance, should be reported. Additional work is needed. A comparison to the current state-of-the-art is also missing.

- Because of the lack of graphene sensor characterization test results, it is unclear how its performance compares with the large body of work already published on this topic. The performance of the sensor as reported (capable of measuring strains between 0.35-0.6%) does not seem impressive. For this specific application, it seems that being able to measure low strains is critical. How does the sensor perform in that regard, and what is the accuracy and resolution?
- The inclusion of previous work on silver nanoparticles does not seem to fit with this manuscript.
- A machine learning method was presented to estimate tire pressure from tire normal load, longitudinal velocity, and the V1 strain measurement parameter. While some of these parameters could be determined in a vehicle, it is unclear how (e.g. tire normal load) could be measured directly. Would this be from the graphene sensors too? What is the novelty of this method?
- The energy harvesting work and results shown is rather simplistic and common, especially when performed using a shaker. Again, the novelty and significance of this is unclear.
- The title of this work suggests that there should be a more well-developed prototype. This study investigated individual pieces of this problem separately and with rather simple methods.

Based on the comments above, the reviewer did not find sufficient novelty and significance for this work to be published in Nature Communications.

Answers to Referee's comments to authors:

We thank the reviewers for their time and effort in reviewing the manuscript. The comments and suggestions have been highly helpful in improving the quality of our study. We have tried our best to address all the comments in the revised submission. The changes are marked in red color in the revised version.

Reviewer #1 (Remarks to the Author):

Overall, the experiments are well-performed and except for some comments that are discussed below, the data and conclusions are reasonable. I would recommend the manuscript for publication in Nature Communications, provided the authors reasonably address the following point.

Thanks for encouraging feedback and recommendation.

Comment: The main concerns about this manuscript is that it is still unclear the proposed strain sensor, energy harvesting, and wireless transmission can withstand the harsh tire operating conditions, namely abrupt very large strain, high cyclic loading, high strain rate (corresponding to the driving speed of 120 km/h), and high temperature during driving. The authors should add some explanation about the limitation of the proposed system so that the reader can understand and utilized the knowledge obtained in this study more effectively.

Response: Thanks for the suggestions. Prior studies have indicated that it is highly unlikely to reach tire temperatures more than 80°C under normal operating conditions for a vehicle, as can be seen in Figure R1.¹ A slight heating was observed for tread area (area in road contact), however, sidewall area slightly became cooler. The tread temperature was found to be less likely affected by the speed.

Figure R1- Tire temperature (Tread and sidewall) versus speed plots.¹

Our experimental data indicates that the sensor can withstand a high cyclic loading and a large strain without getting damaged. Thanks to corrugated sheets of graphene in the 3D printed strain

sensor. With increasing tire rotational speed, stress/strain has been found to decrease due to increase in the acceleration force. The stress (strain) increases slightly at higher speeds ~ 113km/h, however, its value was found to be lower as compared to that at the lower speeds, as shown in Fig.R2. [International Journal of Vehicle Design 65(2/3):270 – 285] With sampling frequency of 10 kHz, we were able to record high resolution data at the desired speed. The frequency of strain rate change (which depends on the speed) for commercial vehicle will be lower than that of collection frequency and thus our measurements cover the extreme scenario.

Figure R2- Calculated stress in the contact load section for rotation rates from 0 to 1200 rpm (Equivalent to 0 to 113 km/h (70.2 Miles/h)). [International Journal of Vehicle Design 65(2/3):270 – 285]

We have conducted temperature stability analysis for the piezoelectric strain harvester. We did not find significant effect on piezoelectric harvester's response in the operational temperature regime (Fig.R3). [Appl. Ener. 232 (2018) 312] This is expected as piezoelectric material has higher Curie temperature compared to the operational range.

Figure R3- Performance of piezoelectric patch at different temperatures. Inset image shows an infrared camera photograph indicating temperature distribution. [Appl. Ener. 232 (2018) 312]

The reduced graphene has good thermal stability up to 500°C [RSC Adv., 2018, 8, 30412], as shown in Fig. R4. This provides robustness needed for rGO based sensors at temperatures less than 80°C. Based on above experimental data, it is clear that the printed rGO sensor and piezoelectric energy harvester can easily withstand harsh tire environment. Both possess the desired flexibility and temperature capability needed for operations over the expected driving speeds.

Figure R4- TGA plot of reduced graphene oxide. [RSC Adv., 2018, 8, 30412]

Reviewer #2 (Remarks to the Author):

Comments:

Overall comments

This paper has several interesting components on material, manufacturing, modeling, simulation, and machine learning for the realization of self-powered sensors for tires in order to contribute to the driverless car effort. It is timely and the paper posed important problem. The main point emphasized here was the ability to 3D print graphene as a strain sensor and self-powering (combined effort of the two aspects) for the tire and evaluation of the system in a field test. Authors have used aerosol printing technique and they have prepared their own ink in order to print the graphene, with the detail chemistry provided in the manuscript. It was also stated as

cost-effective

solution.

Response: We highly appreciate reviewer's encouraging feedback.

Comment: There are some areas authors might consider to improve the technical detail as well as the literature, to make the paper compelling as well as address the knowledge gap in the literature in convincing manner. For example, some closely related papers on graphene sensing and fabrication are listed below.

[1] Cheng, Yin, Ranran Wang, Jing Sun, and Lian Gao. "A stretchable and highly sensitive graphene-based fiber for sensing tensile strain, bending, and torsion." *Advanced materials* 27, no. 45 (2015): 7365-7371.

[2] Yan, Chaoyi, Jiangxin Wang, Wenbin Kang, Mengqi Cui, Xu Wang, Ce Yao Foo, Kenji Jianzhi Chee, and Pooi See Lee. "Highly stretchable piezoresistive graphene–nanocellulose nanopaper for strain sensors." *Advanced materials* 26, no. 13 (2014): 2022-2027.

Related works in 3D printing of graphene.

[3] Burela, Ramesh Gupta, Jagath Narayana Kamineni, and Dineshkumar Harursampath. "Multifunctional polymer composites for 3D and 4D printing." In *3D and 4D Printing of Polymer Nanocomposite Materials*, pp. 231-257. Elsevier, 2020.

[4] Kamran, Urooj, Young-Jung Heo, Ji Won Lee, and Soo-Jin Park. "Functionalized carbon materials for electronic devices: a review." *Micromachines* 10, no. 4 (2019): 234.

[5] Toyserkani, Ehsan, and Elahe Jabari. "Method and apparatus for aerosol-based three-dimensional (3D) printing of flexible graphene electronic devices." U.S. Patent 10,400,119, issued September 3, 2019.

[6] Kalsoom, Umme, Pavel N. Nesterenko, and Brett Paull. "Recent developments in 3D printable composite materials." RSC advances 6, no. 65 (2016): 60355-60371.

[7] Wei, Xiaojun, Dong Li, Wei Jiang, Zheming Gu, Xiaojuan Wang, Zengxing Zhang, and Zhengzong Sun. "3D printable graphene composite." Scientific reports 5 (2015): 11181.

Response: Thanks for suggestion. Above mentioned references have been added to the manuscript.

Comment: The paper is great but there are some issues that are not clear to this reviewer. The self-powered aspect is very important and authors showed using a piezoelectric system. However, how is the graphene base sensor (piezoresistive material) connected with the flexible piezoelectric energy harvester (PVDF)? Some information such as size of the harvester is also missing.

Response: We appreciate reviewer's suggestion. The flexible piezoelectric strain energy harvester was developed to power wireless data transfer electronics that transmit measured varying resistance across the sensor due to change in strain. The piezoelectric harvester was mounted on a piece of tire to continuously harvest strain energy and the generated electrical energy was stored in a capacitor. Once the capacitor is fully charged, this stored energy was used to power the wireless data transfer electronics.

On page 12, 2nd para, line 4, the text has been modified by adding suggested information.

“.....tire) and a corresponding mobile app. Fig. 8(a) shows the set up for energy generation using a piezoelectric patch (3cmx7cm) mounted on a piece of tire, which was excited with a shaker at a selected frequency and amplitude. The measurement frequencies were chosen based on the certain angular velocities of a tire with a given diameter. The energy generated was stored in a capacitor (5 μ F) for powering the wireless sensor, which was mounted on a tire of a.....”

Comment: Some other comments on the theory. It looks like Fig.4 is simulation results. If so, please put simulation results in the caption of Fig.4.

Response: Thanks for mentioning this. The figure caption has been modified accordingly.

On Page 23, The caption of fig 4 has been modified accordingly:

“.....**Figure 4.** (a) The strain as a function of time for three speeds is estimated using eq (5) for a load of 3000 N and pressure of 28 psi. It can be observed that the amplitude of the strain remains the same with speed. (b) The simulated strain in the tire as a function of time for three loads is shown for a speed of 5 mph, and pressure of 28 psi. (c) The simulation results for strain as a function of time for three pressures at a speed of 5 mph and a load of 3000 N. (d) The simulated radial displacement as a function of angular position for three tire pressures for a load of 3000 N and speed of 5 mph. It can be observed that the displacement and hence the strain increases with decrease in the tire pressure for a given load. (e) A representative plot of the simulation results for tire deformation (m) at any given speed. The displacement shown corresponds to that at 5 mph, 3000 N force and 28 psi pressure with a five times amplification.....”

Comment: How are the material properties of the 3D printed graphene included in the simulation? Not clear. It looks like Eq.5 doesn't have the material property of graphene. Please comment on the manuscript.

Response: Thanks for pointing this out. The text has been modified for clarity:

On page9, Line 13,

“.....The in-plane strain given by eqn (5) was derived by assuming the inextensible condition. Furthermore, it is assumed that the graphene sensor weight and stiffness have negligible effect on the tire dynamics. A complete transfer of strain from tire to the graphene sensor is assumed due to the negligible damping occurring at the interface between tire and sensor. Thus, the strain generated due to tire interaction with road is independent of the material properties of the graphene sensor. The strain estimated by eq (5) is taken to be the strain experienced by the graphene sensor. The height z_0 from the neutral layer towards the inner surface of the tire ring corresponded to the sensor mounting location.....”

Comment: Some areas need clarification or discussion. For example, In Fig. 3(e)-(g), why the outputs have different magnitude of bias? This is not discussed.

Response: Thanks for pointing this out. The sentence has been modified for clarity.

On page 7, 2nd para, line 9,

“.....tire pressures (20, 25, 28 psi). Please note the change in voltage waveform due to change in various factors. The bias in the voltage output at each tire pressure was attributed to the shift in the mean resistance of sensor with the ambient pressure. For a given normal load and tire pressure of 28 psi, the peak in the voltage output was found”

Comment: On the modeling, I think the purpose of the modeling is to compare with the experiment. The best way to show is the error between the simulation model and the experimental results in a plot (if possible). If not, in a text, explaining error magnitudes.

Response: Thanks for suggestion. We have provided comparative analysis of modeling and experimental results. The text has been modified accordingly.

On page 10, 1st para, line 13,

“.....A similar observation could be noted in the radial displacement of the tire. In experiments, the voltage is measured across an equivalent resistance connected in series with the graphene sensor and a DC power supply. The comparative analysis of modeling and experimental results (Fig. S9) clearly indicate that the simulated strain response follows the same trend of experimental results shown in Fig 3 (e). As shown in Fig. 4(d), the peak disp.....”

In supplementary information, following paragraphs and Fig. S9 are added:

As shown in Fig. 3(c), in each rotation, tire has compressive strain (zone P or Q) and tensile strain (zone R). For comparing experimental and modeling results, we define the ratio (K) as:

$$K = \frac{\text{Compressive zone amplitude (C)}}{\text{Tensile zone amplitude (T)}}$$

We calculated the experimental K value and compared this ratio with our modeling results. Fig. S9 shows the experimental and modeling results on the same x-axis. We found that $K_{\text{Experiment}} = 0.25$ and $K_{\text{Model}} = 0.174$. This matching is quite significant considering that the data was collected in real environment in contrast to controlled lab environment.

Figure S9. Simulation results for the strain and experimental results (in voltage) when the tire pressure is 28 psi for a 4000 N load at 5 mph. Updown arrows show the tensile and compressive strain amplitude of modeling and experimental results.

Comment: Another question: From where Eq.1 came from? Where is Eq.5 used? These issues need clarification.

Response: Thanks for pointing this out. The text has been modified for clarity.

On page 8, 2nd para, line 7:

“.....The governing equation of motion for a rotating inextensible tire with radius r , width b , thickness t , and internal pressure p is derived using the Hamilton’s principle². The equation of motion for a non-rotating tire to determine the natural frequencies is obtained by ignoring the tire rotation and tire loading as:

$$\frac{EI}{r^4}(v^6 + 2v^4 + v'') - \frac{pb}{r}(v^4 + v'') + k_w v'' - k_v v - \rho b t(\ddot{v} - \ddot{v}'') = 0. \quad (1)$$

The frequencies were estimated by assuming a series solution for the in-plane deflection, v”

On page 9, line 16

“.....Thus, the strain generated due to road interactions is independent of the material properties of the graphene sensor. The strain estimated by eq (5) is taken to be the strain experienced by the graphene sensor. The height z_0 from the neutral layer towards the inner surface of the tire ring corresponded to the sensor mounting location.....”

Comment: It was shown that the 3D printed graphene can measure strain 3500 to 6000 microstrin, and it has four layers. What was the thickness of each layer and how about the total 3D printed part thickness?

Response: Thanks for mentioning this. We have included thickness information in the text.

On page 6, 1st para, line8

“.....and viscosity, we were able to achieve graphene oxide sensors with homogenous microstructure (Fig. 2(d)-(g)). The final graphene based printed sensor was obtained by 5 number of passes with total thickness of approximately 10 μ m.....”

Comment:

Machine learning: The results in Fig.5 looks great. It can be easily seen the effects of load and pressure. It was stated that “experimental data of 1023 tire revolutions collected using the portable tire-testing trailer” What was the data size, training time, conditions, and hyper parameters?

Response: Thanks for your comment, to show the effectiveness of the printed sensors in estimating some of the tire related parameters. Experiments were designed with three different longitudinal speeds (5, 10, 20 mph), three different normal loads (2000, 3000, 4000 N) on the

tires having three different inflation pressures (20, 25, 28 psi). The experimental data for 1023 tire revolutions from combination of different testing condition collected using the portable tire testing trailer was used to develop the machine learning algorithm. From this dataset, 70 percent of the data was used for training, 15 percent for validation and 15 percent for testing. A two-layer feed-forward Neural Network (NN) with sigmoid hidden neurons that uses ten neurons in its hidden layer was used. The Levenberg-Marquardt optimization algorithm was used for training (this makes the training time very short- less than a second in this case) and Mean Squared Error was used to evaluate the performance. The performance of the algorithm is shown in Figure 6.

Comment: Lastly, the cost-effective solution needs some substantial information, quantifying will make it stronger.

Response: Thanks for suggestions. We have quantified the cost of 3D printed sensor.

One page 3, 1st para, line 11,

“.....with energy harvesting and secure data transmission are cost-effective solutions for smart tires (Fig.1). The cost of a 3D printed sensor in this work was estimated to be roughly 2.7 cents as described in supplementary information (Table –S1). Graphene-based 3D-printed strain sensors are shown to acquire environmental information and wirelessly.....”

Supplementary information.

Table S3- Sensor cost estimation

Usage amount and price required for rGO synthesis					
Materials	Price (\$) ¹	Quantity	consume	Price (\$)	
Synthetic graphite powder (< 20 um)	70.6	1 kg	1 g	0.07	
Sulfuric acid, 98%	85.7	2.5 L	120 mL	4.11	
Phosphoric acid solution, 85 wt.%	153	2.5 L	15 mL	0.92	
Potassium permanganate ACS reagent, ≥99.0%	275	2500 g	6 g	0.66	
Hydrogen peroxide solution ACS reagent, 30%	98.5	500 mL	3 mL	0.59	
Hydriodic acid (HI), ACS reagent, 55%	646	1000 mL	3 mL	1.94	
				8.29	per gram of precursor (graphite) (commonly, we say that the yield of rGO is 70%)
				Thus,	11.8 USD/gram of rGO
1. The costs of materials and chemicals are set by the price offered by Sigma-Aldrich Co.					
					The solution was diluted to 0.03 g wt% for making ink
Electricity consumption cost calculation in printing one sensor					100ml Ink cost (Approx)
Time for printing one sensor (h)	0.083				0.355285714 USD
Machine Wattage	2200				
Wh	182.6			One 10x10 mm sensor requires ink	2ml
Machine usage in kWh for one sensor	0.18			Material cost of printing one sensor	0.007105714 USD
Virginia electricity cost	11.1 cents/kWh				
Electricity cost in printing one sensor (cents)	1.998				
	0.01998 USD				
				Total cost of printing one sensor	0.027085714 USD
					2.708571429 Cents

The cost of one printed sensor is around 2.7 cents. The scaling of the processing will further reduce the cost of the sensor due to lower raw material cost.

Comment: Some of the figures are low resolution, less clear and need replacement.

Response: Sorry for low resolution figures. We have increased resolution of figures in the revised manuscript.

Comment: Overall, this paper presented a 3D printed graphene sensor for tires of autonomous vehicles, which has been validated in a field test showing functionality over a range of parameters such as pressure and normal load. The entire work is a very good example of cyber physical system, as it has data security and communication, sensing, and material innovation, monitoring, and machine learning. The issue of realizing usable sensors like graphene that is reliable and feasible solution for autonomous vehicles is an important problem.

Response: We highly appreciate reviewer's encouraging feedback.

Reviewer #3 (Remarks to the Author):

This paper presents an experimental study that investigated separately the different components that would comprise of a smart tire. A main focus of this study was to present a 3D-printed graphene strain gage. The reviewer found several issues with this manuscript, and the comments are as follows:

Comment: The abstract states that dynamic parameters such as road condition, friction, slip, pressure, and strain distributions are required. This study does not demonstrate that any of these parameters could be directly measured.

Response: Thanks for mentioning this. We have modified the abstract for clarity. The focus of this study is to demonstrate 3D printed cost effective and robust tire strain sensor (as compared to off-the-shelf strain sensors, used in prior tire studies), which can be used to estimate parameters such as friction, slip, pressure, and strain distributions. To demonstrate the effectiveness of the developed sensor, a sampling algorithm (pressure monitoring algorithm) was developed. There are several studies that show the effectiveness of strain sensors, in general, in estimating the tire's overall conditions and tire-road contact parameters. Some of these references are listed below:

X. Yang, O. Olatunbosun, D. Garcia-Pozuelo Ramos, and E. Bolarinwa, "Experimental investigation of tire dynamic strain characteristics for developing strain-based intelligent tire system," SAE Int. J. Passenger Cars Mech. Syst., vol. 6, no. 1, pp. 97–108, Apr. 2013.

X. Yang, O. Olatunbosun, D. Garcia-Pozuelo, and E. Bolarinwa, "FEBased tire loading estimation for developing strain-based intelligent tire system," SAE Technical Paper Series, Apr. 2015.

H. Y. Jo, M. Yeom, J. Lee, K. Park, and J. Oh, "Development of intelligent tire system," SAE Technical Paper Series, Apr. 2013.

R. Matsuzaki and A. Todoroki, “Intelligent tires for improved tire safety based on strain measurements,” in Proc. Health Monitoring of Structural and Biological Systems, San Diego, CA, 2009.

R. Matsuzaki, N. Hiraoka, A. Todoroki, and Y. Mizutanj, “Analysis of applied load estimation using strain for intelligent tires,” JSME Int. J. A, Solid M, vol. 4, no. 10, pp. 1496–1510, 2010

Lee, H., & Taheri, S. (2017). Intelligent tires? A review of tire characterization literature. *IEEE Intelligent Transportation Systems Magazine*, 9(2), 114-135.

Basically, all the strain-based algorithms (using off-the-shelf sensors) for estimating various tire related parameters can be extended to the 3D printed strain sensor as well.

Comment: Complete characterization of the graphene-based sensor is basically nonexistent in this paper. The only result presented was the sensor mounted on a piece of tire and loaded. Sensor properties, as well as their cyclic response and performance, should be reported. Additional work is needed. A comparison to the current state-of-the-art is also missing.

Response: We appreciate your suggestion. We have performed extensive real environment field test on our 3D printed tire sensor. The 3D printed sensor was mounted on a tire that was mounted within a test rig in a trailer. During this testing over a distance of more than 10 miles, the tire sensor provided reliable measurements without any issue. This demonstrates the robust nature of our 3D printed sensor with ability to withstand operational cyclic strain. Table below provides summary of the experiments performed in real environment.

Table S1- Summary of the experiments performed in real environment.

Tire pressure (psi)	Normal load (N)	Speed (MPH)
20	2000	5, 10 and 15

		3000	5, 10 and 15
		4000	5, 10 and 15
25		2000	5, 10 and 15
		3000	5, 10 and 15
		4000	5, 10 and 15
28		2000	5, 10 and 15
		3000	5, 10 and 15
		4000	5, 10 and 15

During real environment field test on the 3D printed sensor, the sensor was subjected to a large number of cycles ~8031. The calculation for the number of cycles is added in supplementary information (Table S2). The stability of sensor under such a large number of cycles in real environment clearly attests durability and practicality of the approach.

Table S2- Calculation of number of cycles of 3D printed sensor during real environment test.

	5 mph	Time difference	10 mph	Time difference	20 mph	Time difference		
2000 N 28 PSI	12:15:22 PM		12:55:12 PM		12:57:25 PM			
	12:51:47 PM	0:36:25	12:56:27 PM	0:01:15	12:59:10 PM	0:01:45		
3000 N 28 PSI	1:01:51 PM		1:10:35 PM		1:16:29 PM			
	1:03:27 PM	0:01:36	1:12:40 PM	0:02:05	1:17:51 PM	0:01:22		
4000 N 28 PSI	1:19:12 PM		1:22:21 PM		1:24:36 PM			
	1:21:38 PM	0:02:25	1:24:13 PM	0:01:52	1:25:57 PM	0:01:21		
2000 N 25 PSI	1:32:45 PM		1:36:49 PM		1:39:58 PM			
	1:35:36 PM	0:02:52	1:38:43 PM	0:01:54	1:41:02 PM	0:01:04		
3000 N 25 PSI	1:44:01 PM		1:58:40 PM		2:05:53 PM			
	1:58:14 PM	0:14:13	2:04:40 PM	0:06:00	2:06:52 PM	0:00:59		
4000 N 25 PSI	2:08:37 PM		2:14:57 PM		2:16:57 PM			
	2:13:44 PM	0:05:06	2:15:58 PM	0:01:01	2:18:11 PM	0:01:14		
2000 N 20 PSI	2:23:48 PM		2:27:09 PM		2:29:25 PM			
	2:25:49 PM	0:02:02	2:28:16 PM	0:01:07	2:30:33 PM	0:01:09		
3000 N 20 PSI	2:32:09 PM		2:40:59 PM		2:43:29 PM			
	2:33:59 PM	0:01:50	2:42:09 PM	0:01:10	2:44:29 PM	0:01:01		
4000 N 20 PSI	2:46:18 PM		2:48:58 PM		2:51:19 PM			
	2:47:23 PM	0:01:05	2:50:04 PM	0:01:05	2:52:14 PM	0:00:54		
Total running time		1:07:34		0:17:29		0:10:48		
Wheel perimeter (Inch)		96		96		96		
Running distance (miles)		5.63		2.91		3.60		Grand total:
# cycles		3723		1927		2380		8031

The experimental tests in real environment are provided in main text (Fig 3(e)-(g)) and supplementary information (Fig. S7). The experiments performed in the controlled lab environment are provided in Fig. 2(i)-(l). We have performed extensive materials characterization of the 3D printed sensor including Raman, TEM, AFM, and XRD etc. Similar

to the scaling of other material-based technology for commercial application, we think that 3D printed sensor technology will require additional testing, which would be the focus of the future work.

Comment: Because of the lack of graphene sensor characterization test results, it is unclear how its performance compares with the large body of work already published on this topic. The performance of the sensor as reported (capable of measuring strains between 0.35-0.6%) does not seem impressive. For this specific application, it seems that being able to measure low strains is critical. How does the sensor perform in that regard, and what is the accuracy and resolution?

Response: Thank you for the comment. Generally, off-the-shelf strain sensors used in prior strain-based intelligent tire studies have shown the capability of measuring strain up to 0.40-0.50%. In comparison, the 3D printed sensor presented in this study is capable of measuring the strain up to 0.7%. Fig 2(l) shows the performance of the 3D printed sensor under bending. It can be seen here that the change in resistance versus strain is almost linear for the strain below 0.7% that shows the effectiveness of the printed sensor in measuring the deformation below this strain magnitude. This covers the strain range required for the tire-road interaction under the normal driving conditions.

Comment: The inclusion of previous work on silver nanoparticles does not seem to fit with this manuscript.

Response: Thanks for suggestions. We have moved the information related to printing of silver nanoparticles in supplementary information.

Comment: A machine learning method was presented to estimate tire pressure from tire normal load, longitudinal velocity, and the V1 strain measurement parameter. While some of these parameters could be determined in a vehicle, it is unclear how (e.g. tire normal load) could be measured directly. Would this be from the graphene sensors too? What is the novelty of this method?

Response: Thank you for the comment. The machine learning algorithm was presented in this study as an example of the effectiveness of the printed sensor data in estimating the tire-related parameters. The normal load applied on the tire was controlled and measured using a portable tire test trailer, however, there are many other studies that have presented different algorithms to estimate the normal load using various types of sensors such as accelerometers and piezoresistive sensors (for the piezoresistive sensors, the time deference between the peaks of the time derivative of the signal, shown in Figure 5, can be used to estimate the normal load). This work, for the first time, successfully demonstrates printed high performance and cost-effective graphene based strain sensors and its integration with an actual tire for testing in real environment. Please note that during the 10 miles of road test data collection, the sensor was functional and provided tire strain and pressure information. Considering the impact of this work, several companies affiliated with the CENTIRE (An NSF funded tire research center at Virginia Tech) have expressed interested in testing these sensors. Some of the prior studies from our team members in estimating tire related parameters using different kinds of commercial sensor are listed below:

Lee, H., & Taheri, S. (2017). Intelligent tires? A review of tire characterization literature. *IEEE Intelligent Transportation Systems Magazine*, 9(2), 114-135.

Lee, H., Kim, M. T., & Taheri, S. (2018). Estimation of Tire–Road Contact Features Using Strain-Based Intelligent Tire. *Tire Science And Technology*, 46(4), 276-293.

Khaleghian, S., Ghasemalizadeh, O., Taheri, S., & Flintsch, G. (2019). A Combination of Intelligent Tire and Vehicle Dynamic Based Algorithm to Estimate the Tire-Road Friction. *SAE International Journal of Passenger Cars-Mechanical Systems*, 12(2), 81-98.

Behroozinia, P., Khaleghian, S., Taheri, S., & Mirzaeifar, R. (2020). An investigation towards intelligent tyres using finite element analysis. *International Journal of Pavement Engineering*, 21(3), 311-321.

Behroozinia, P., Khaleghian, S., Taheri, S., & Mirzaeifar, R. (2019). Damage diagnosis in intelligent tires using time-domain and frequency-domain analysis. *Mechanics Based Design of Structures and Machines*, 47(1), 54-66.

This study paves the pathway for using cost-effective 3D printed tire strain sensors (that can be directly printed on the tire inner liner) to estimate the tire-related parameters. It has strong potential to overcome issues associated with off-the-shelf sensors (such as cost, robustness, need for external power, etc).

Comment: The energy harvesting work and results shown is rather simplistic and common, especially when performed using a shaker. Again, the novelty and significance of this is unclear.

Response: The focus of this work is to demonstrate an integrated system – cost effective scalable 3D printed tire strain sensors combined with piezoelectric energy harvester based wireless data transfer and neural network based data analytics (data collection, data transfer and data analysis). The flexible piezoelectric tire strain energy harvester is developed to power the

wireless data transfer electronics and measure resistance change across the 3D printed sensor without external power source. To demonstrate the feasibility of the proposed energy harvesting system, a piezoelectric harvester was mounted on tire to harvest strain energy and charge capacitor (5 μ F).

Comment: The title of this work suggests that there should be a more well-developed prototype. This study investigated individual pieces of this problem separately and with rather simple methods.

Based on the comments above, the reviewer did not find sufficient novelty and significance for this work to be published in Nature Communications.

Response: As mentioned above, the focus of this work was to demonstrate an integrated system – cost effective scalable 3D printed tire strain sensors combined with piezoelectric energy harvester based wireless data transfer and neural network based data analytics (data collection, data transfer and data analysis). At each stage, we have provided fundamental advancement in facilitating the integration – novel 3D printed graphene sensors to piezoelectric patches for energy harvesting to mathematical algorithm for interpreting the results.

We appreciate reviewers' time and efforts in reviewing this manuscript. We have performed extensive analysis and clearly demonstrated practical feasibility of our 3D printed sensor by integrating sensors in an actual tire and conducting real environment field tests. We believe our study provides a holistic approach and marks a crucial step towards developing smart tires for next generation of autonomous vehicles.

References:

1. Wilburn DK. A temperature study of pneumatic tires during highway operation.). ARMY TANK-AUTOMOTIVE COMMAND WARREN MI (1972).
2. Wei YT, Nasdala L, Rothert H. Analysis of forced transient response for rotating tires using REF models. *Journal of Sound and Vibration* **320**, 145-162 (2009).

Reviewer #1 (Remarks to the Author):

The authors have made all the necessary changes from myself, and I also appreciated the input from the other reviewer of which I felt also contributed greatly to the journal.

Reviewer #2 (Remarks to the Author):

The authors have address all the questions raised by this reviewer. They also added new results about the 3D printed sensors.

Reviewer #3 (Remarks to the Author):

The authors have addressed most of the reviewer's comments/concerns.